# Frequency and Consequences of Cervical Lymph Node Overstaging in Head and Neck Carcinoma

**DOI:** 10.3390/diagnostics12061377

**Published:** 2022-06-02

**Authors:** Volker Hans Schartinger, Daniel Dejaco, Natalie Fischer, Anna Lettenbichler-Haug, Maria Anegg, Matthias Santer, Joachim Schmutzhard, Barbara Kofler, Samuel Vorbach, Gerlig Widmann, Herbert Riechelmann

**Affiliations:** 1Department of Otorhinolaryngology-Head and Neck Surgery, Medical University of Innsbruck, Anichstr. 35, 6020 Innsbruck, Austria; volker.schartinger@i-med.ac.at (V.H.S.); natalie.fischer@tirol-kliniken.at (N.F.); praxis@hno-am-dez.at (A.L.-H.); maria.anegg@student.i-med.ac.at (M.A.); matthias.santer@tirol-kliniken.at (M.S.); joachim.schmutzhard@i-med.ac.at (J.S.); ba.kofler@tirol-kliniken.at (B.K.); herbert.riechelmann@i-med.ac.at (H.R.); 2Department of Radiation-Oncology, Medical University of Innsbruck, Anichstr. 35, 6020 Innsbruck, Austria; samuel.vorbach@tirol-kliniken.at; 3Department of Radiology, Medical University of Innsbruck, Anichstr. 35, 6020 Innsbruck, Austria; gerlig.widmann@tirol-kliniken.at

**Keywords:** head and neck neoplasm, neck dissection, tumor staging, computed tomography

## Abstract

Clinical lymph node staging in head and neck carcinoma (HNC) is fraught with uncertainties. Established clinical algorithms are available for the problem of occult cervical metastases. Much less is known about clinical lymph node overstaging. We identified HNC patients clinically classified as lymph node positive (cN+), in whom surgical neck dissection (ND) specimens were histopathologically negative (pN0) and in addition the subgroup, in whom an originally planned postoperative radiotherapy (PORT) was omitted. We compared these patients with surgically treated patients with clinically and histopathologically negative neck (cN0/pN0), who had received selective ND. Using a fuzzy matching algorithm, we identified patients with closely similar patient and disease characteristics, who had received primary definitive radiotherapy (RT) with or without systemic therapy (RT ± ST). Of the 980 patients with HNC, 292 received a ND as part of primary treatment. In 128/292 patients with cN0 neck, ND was elective, and in 164 patients with clinically positive neck (cN+), ND was therapeutic. In 43/164 cN+ patients, ND was histopathologically negative (cN+/pN−). In 24 of these, initially planned PORT was omitted. Overall, survival did not differ from the cN0/pN0 and primary RT ± ST control groups. However, more RT ± ST patients had functional problems with nutrition (*p* = 0.002). Based on these data, it can be estimated that lymph node overstaging is 26% (95% CI: 20% to 34%). In 15% (95% CI: 10% to 21%) of surgically treated cN+ HNC patients, treatment can be de-escalated without the affection of survival.

## 1. Introduction

In recent years, we have seen several patients with head and neck cancer (HNC) with clinically positive cervical lymph nodes (cN+) who were originally scheduled for surgical resection, therapeutic neck dissection (ND) and postoperative radiotherapy (PORT) by the Interdisciplinary Head and Neck Tumor Board (ITB). Unexpectedly, the resected neck specimens had negative histopathology (pN0). The lymph node status was downstaged from N+ to N0 (Figure 1). Some of these patients had a primary T1–2 tumor, and the disease was downgraded from a locally advanced stage (III/IV) to a localized stage (I/II). If the primary tumor was resected with clear margins (R0), PORT was often omitted. In these patients, initial clinical lymph node overstaging was corrected, treatment was de-escalated and treatment-related morbidity was possibly reduced. If a combined radiotherapy with systemic therapy (RT + ST) had been performed primarily instead of surgical therapy in these patients, overstaging would have gone unnoticed. Based on these observations, we had several questions:How often did histopathological examination of ND specimens show pN0 status in surgically treated HNC patients clinically classified as lymph node positive (cN+)?How often has this downstaging of lymph nodes led to a de-escalation of the original treatment plan from multimodal to unimodal treatment? In other words, how often was an originally planned PORT avoided?Were there differences in overall survival in patients who were downgraded in this manner and who did not receive PORT, and patients who were a priori classified as cN0 and had undergone single modality tumor resection with elective ND?How did overall survival and functional outcome in these downstaged patients compare to statistically matched patients receiving definitive radiotherapy (RT) or RT with systemic therapy (RT ± ST)?

In a retrospective registry study, we have tried to find some answers to these questions. The basic question was: is neck dissection both a diagnostic procedure to detect occult cervical metastases and also a diagnostic procedure to exclude histopathologically positive neck nodes and allow treatment de-intensification in some HNC patients?

## 2. Materials and Methods

### 2.1. Data Assessment

Patients with incident HNC treated at the Department of Otolaryngology, Medical University of Innsbruck between 2008 and 2018 were identified in the Clinical Head and Neck Cancer Registry. Clinical TNM staging was carried out during the Interdisciplinary Tumor Board (ITB) meetings of the Medical University of Innsbruck, jointly by head and neck surgeons, radiologists, radiotherapists, oncologists and pathologists. ITB staging for each individual patient was based on clinical findings, examination under anesthesia with endoscopy and available imagery [1]. A high-resolution CT scan with contrast medium of the head, neck and trunk is the minimum imaging requirement to present a patient in our ITB.

The ITB then submitted individual treatment recommendations for each patient in close accordance to current Clinical Practice Guidelines in Oncology of the American National Comprehensive Cancer Network. In general, clinical T1–2 and cN0 disease (UICC stage I/II) were treated either by surgery alone or by radiotherapy alone. Depending on various factors including the primary tumor site, tumor size and depth of invasion, elective, usually selective neck dissection (SND) was performed in surgically treated cN0 patients [2]. Patients with clinical T3–4 tumors or clinically positive cervical lymph nodes (cN+) were staged as advanced disease (UICC stage III/IV) except for some p16-positive oropharyngeal carcinomas. In patients with advanced disease, tumor resection including unilateral or bilateral therapeutic ND was advocated as surgical treatment. In patients with cN+ neck, a comprehensive neck dissection was usually recommended and in some patients with a small suspicious node a SND [3]. In patients with advanced disease, surgery was usually followed by PORT or by RT + ST in the case of adverse features [4]. As an alternative to surgery and PORT, RT + ST was frequently recommended in patients with advanced disease [5].

### 2.2. Study Population

Inclusion criteria were previously untreated patients with histologically confirmed incident HNC of any clinical UICC Stage from any site except thyroid, parathyroid gland and unknown primary, who underwent unilateral or bilateral neck dissections during initial treatment. Disease was staged according to the UICC TNM staging system (TNM, 7th edition, 2010) for the patients diagnosed before 2017 and to UICC TNM staging system (TNM, 8th edition, 2017) afterwards [6,7]. For each patient, clinical TNM stage (cTcNcM) was defined within weekly Multidisciplinary Tumor Board sessions by the ITB members based on multidisciplinary clinical and radiological judgement. The review board of the Medical University of Innsbruck approved the study (AN2015-0140).

### 2.3. Imaging Modalities

Contrast-enhanced CT according to the standard head and neck imaging protocols at the Department of Radiology, Medical University of Innsbruck, was the basic imaging modality and available in all patients. A GE-medical Systems Light speed VCT or Light speed 16 CT scanner (GE Medical, Vienna, Austria) was used. The scan area ranged from the frontal sinus to the upper mediastinum. Slices were reconstructed with 2 mm thickness and additional sagittal and coronal images were obtained. As contrast medium, Jopamiro 370 (Bracco Austria GmbH, Vienna, Austria) was administered intravenously adjusted to the patient’s bodyweight (2 mL per kg bodyweight up to 120 mL maximum dose). The images were exported in Digital Imaging and Communications in Medicine (DICOM) format using IMPAX EE (Agfa HealthCare, Bonn, Germany) Picture Archiving and Communication System (PACS). CT scans were evaluated by a specialist in head and neck radiology. CT criteria for malignancy were a round shape with or without central necrosis and/or a minimal axial diameter greater than 10 mm [8,9]. Frequently, additional imaging modalities including ultrasound, MRI and/or PET-CT were available and were considered during clinical staging.

### 2.4. Assessment of Cervical Lymph Node Size

For the measurement of orthogonal lymph node diameters in millimeters (mm), axial and coronal CT images in diagnostic contrast-enhanced CT scans were used [10]. Maximum diameters were assessed in anterior–posterior, medio-lateral and craniocaudal directions using a standard visualization software (PACS, Cerner, Kansas City, MO, USA). The maximal short axis diameter (SAD), i.e., the axis perpendicular to the longest axis of the lymph node, of the largest cervical lymph node was assessed in accordance with the criteria by van den Brekel et al. [8,9].

### 2.5. Histopathologic Examination

Neck specimens were orientated by the surgeon and placed in formalin. Lymph nodes >3 mm were identified by inspection and palpation. Each discrete node was dissected out with attached pericapsular adipose tissue. Larger nodes were sliced. In case of obvious metastatic tumor, the half/slice with the more extensive tumor was processed together with the perinodal tissues to show the extent of extranodal extension. Hematoxylin- and eosin-stained sections from each block were cut at 4 µm thickness and used for histopathological assessment [11]. A minimum nodal yield of 10 lymph nodes in RND/mRND or 6 lymph nodes in elective SND was considered sufficient for pN-staging, respectively. Usually, the nodal yield was substantially higher.

### 2.6. Assessment of Treatment Functional Outcome

For in-house quality control at the Department of Otorhinolaryngology, Medical University of Innsbruck, a head and neck cancer functional integrity scale has been developed. The physician asks the information from the patient in a brief interview and crosshatches the answers in a response grid. It covers the 6 functional domains nutrition, breathing, speech, pain, mood and neck and shoulder mobility. For each domain, a set of 5 answer options is given, which records the extent of the functional restriction on an ordinal scale based on objective criteria. It ranges from worst functional outcome to normal function. It usually takes less than 2 min to fill in the form. For outcome evaluation, the number of patients reporting each functional status for the 6 functional domains is provided.

### 2.7. Data Analysis

For categorical data, frequency tables are reported and for continuous data averages and standard deviations, unless otherwise specified. To calculate 95% confidence intervals of proportions, the Wilson method with continuity correction was used [12]. Case–control matching was performed using the IBM SPSS Python extension for fuzzy matching [13]. Criteria for matching included gender, age and ASA score as a simple indicator of comorbidity, tumor site, clinical T and N stage and whether it was a p16-positive oropharyngeal carcinoma. The Kaplan–Meier method with log-rank test was used to compare overall survival. The median follow-up was calculated according to Schemper and co-authors [14]. To compare the functional results, the number of patients per functional integrity scale value and functional domain was tabulated for cases and controls and tested with the Kruskal–Wallis test for singly ordered RxC contingency tables with the “Exact” option [15]. The calculations were performed with IBM SPSS 25 (IBM, Austria, Vienna) and StatXaxt Version 8 (Cytel Inc., Cambridge, MA, USA).

## 3. Results

### 3.1. Patient and Disease Characteristics

Between 2008 und 2018, 980 patients with incident HNC complying with the inclusion criteria were treated at the Department of Otorhinolaryngology—Head and Neck Surgery, Medical University of Innsbruck. Of these, 292 patients with incident HNC had received a ND as part of primary treatment. The majority were men (*n* = 227) and the mean age was 61 years (range 29 to 89 years). The most common tumor site was the oropharynx followed by oral cavity, larynx and other sites (Table 1). In 128/292 patients with cN0 neck, ND was elective and in 164/292 patients, cervical lymph nodes had been classified as clinically positive by the ITB and ND was by therapeutic intent.

### 3.2. Frequency of Pn0 Status in Clinically Positive Lymph Node Patients

Histopathological examination revealed negative lymph nodes in 43/164 patients (26%; 95% CI: 20% to 34%), in whom the ITB had assumed positive lymph nodes based on available clinical data and imagery (cN+). Clinical and lymph node characteristics of patients with clinically positive neck specimens grouped by positive (*n* = 121) and negative (*n* = 43) histopathologic results of ND specimens are provided in Table 2.

### 3.3. Frequency of Treatment De-Escalation

Histopathological lymph node downstaging led to treatment de-escalation in 24/43 patients, i.e., the initially planned PORT was omitted. This corresponds to 15% (95% CI: 10% to 21%) of the 164 patients with clinically positive lymph nodes (cN+). In total, 24 out of the 43 downstaged patients had T1–2 primaries and free resection margins. However, the other 19 of the 43 downstaged patients received PORT despite a histopathologically negative neck specimens. In 12 of these, the primary tumor was at least T3. Another four patients with T1–2 primaries had narrow resection margins or dysplasia in the resection margin, and three patients had an unfavorable primary tumor localization at the edge of two adjacent UICC tumor sites.

### 3.4. Further Course of Disease and Survival in Patients Following Treatment De-Escalation

The mean follow-up of 24 patients with originally planned PORT who were finally treated with surgery alone was 35.5 ± 26.6 months. One patient died one month postoperatively without evident disease because of a peritonitis associated with an acquired immune deficiency syndrome. Recurrent disease was observed in five patients. Of these, two developed a cervical lymph node metastasis and three patients a recurrence at the primary site. In these five patients, salvage treatment was initiated, and four of them were disease free at the end of the observation period. One patient died after one year.

A sound comparison group to the 24 downgraded patients with treatment de-escalation (cases) are patients staged as cN0 by the ITB and who were scheduled for primary surgical resection with selective neck dissection without PORT (cN0-controls). We identified 72 patients in our tumor registry who met these conditions. Except for clinical lymph node status, clinical characteristics of cases and controls were similar (data not shown). Overall, the survival of the cases and controls did not differ significantly (Figure 2; log rank *p* = 0.74).

### 3.5. Survival and Functional Outcome in Surgically Treated Cn+ Patients with Treatment De-Escalation and Matching Cn+ Patients Treated with RT ± ST

RT ± ST is a treatment alternative to surgical treatment and PORT in HNC patients with clinically positive lymph nodes [5]. Especially in organ preservation setting and with an unacceptable risk of highly impaired functional outcome RT ± ST is offered to patients in our institution.

In patients treated with RT ± ST, it goes unnoticed if their cervical lymph nodes are pathologically negative and the treatment cannot be adjusted accordingly. Using SPSS case control matching with the fuzzy extension command [13], we identified 21 matched pairs, i.e., 21 surgical cases with treatment de-escalation and 21 RT ± ST controls. Sex, age, tumor site, T-stage, N-Stage and p16 positivity in the case of oropharyngeal carcinoma and the American Society of Anesthesiologist score as a simple comorbidity indicator served as the matching parameters [16,17]. The matching 21 surgical cases and the 21 RT ± ST had similar patient and disease characteristics (Table 3). Overall, the survival of the cases and controls did not differ significantly (log rank *p* = 0.81; Figure 3).

Functional integrity was better in the surgically treated cN+ patients with treatment de-escalation. Significantly more surgical patients reported better nutritional function than matched patients with RT ± ST (*p* = 0.002), while the other outcome parameters showed no differences (Table 4).

## 4. Discussion

Clinical staging is an essential part of treatment planning for head and neck cancer. It is usually based on clinical examination and imaging. One key question is whether the cervical lymph nodes are affected. Despite extensive research on the sensitivity and specificity of different imaging modalities [18], this question can often not be definitely answered. Clinical lymph node staging based on all available clinical information and imagery by a multidisciplinary team including experienced radiologists and clinicians is considered the best option and is the reference in the current study. However, even clinical staging by a multidisciplinary team is fraught with uncertainties. For the problem of occult cervical metastases, established clinical algorithms are available [19]. On the other hand, there is also the problem of clinical overstaging of cervical lymph nodes, which may lead to unnecessarily aggressive therapy.

This problem has hardly been investigated and it is methodologically difficult to address. We chose a step-by-step approach to investigate this problem. First, we identified patients in our clinical tumor registry, who were clinically cervical lymph node positive as judged by the ITB (cN+) and for whom surgical therapy and PORT was planned (*n* = 164). From these, we selected those patients in whom neck dissection unexpectedly resulted in a pathologically negative cervical lymph node status (pN0). This was the case in 43/164 patients. Except for one patient with p16+ oropharyngeal cancer, this resulted in a downstaging from UICC stage III or IV to stage I or II for patients with a primary tumor <T3. If pT1–2, pN0-patients also had free resection margins, PORT was omitted. This was the case in 24/164 cN+ patients. Instead of an initially planned multimodality therapy, a single modality therapy was performed. Single modality treatment was associated with lower morbidity than multimodality treatment [20].

To evaluate the further course of these 24 patients, we selected patients with a clinical cT1–2 cN0 stage, for whom the ITB had a priori planned tumor resection and elective neck dissection without PORT. In this group, 72 patients had clear resection margins, negative neck specimens, did not receive PORT and served as the control group. In a Kaplan–Meier analysis, we did not observe a survival difference between these two groups (*p* = 0.74). We therefore assume that these 24 downstaged patients had not suffered any survival disadvantages by omitting the PORT.

Cervical lymph node characteristics of two patient groups, namely cN+/pN0 and cN+/pN+, show that an increase in specificity by sharpening criteria for positivity would lead to a comparable loss of sensitivity (Table 2). In fact, 44% of the false positive LNs were 6–10 mm and 44% 11–15 mm. In contrast, 16% of the true positive LNs were 6–10 mm and 35% 11–15 mm. In several patients, PET-CT was not available, and it may be that routine PET-CT would improve diagnostic accuracy of the ITB. However, PET-CT could not eliminate overstaging. Frequently, PET-CT is not readily available and may increase time to start of treatment [21], especially when false positive tracer uptake in other body regions than head and neck is further investigated. Fine-needle aspiration cytology may also increase diagnostic accuracy, but its high sensitivity and specificity is associated with palpable head and neck masses, not with small lesions [22]. So, it might be that diagnostic accuracy of our ITB can be increased to some extent, but with high costs and probably only to a minor degree. The authors are aware that this unimodal lymph node staging together with the retrospective study design is one of the weaknesses of this study. On the other one might advocate that the underlying concept is worldwide applicable and comparable. We feel that a design reduced to readily available technical standards is a strength of the study at the same time.

Next, we identified non-surgically treated controls with patient and disease characteristics similar to the 24 downstaged patients. These controls had received a RT ± ST which does not allow us to detect potential lymph node overstaging and de-escalate the initially planned treatment. These control patients were identified with the SPSS Fuzzy Matching Algorithm [13]. Fuzzy matching helps to identify two data elements (e.g., T-stage) that are similar but not the same (e.g., cT3 and cT4) [23]. In contrast to traditional matching, where data were either exactly matched or not (cT3 ≠ cT4), the simplest application of fuzzy matching numerically ranks the matching degree based on the dimensions of the data at hand (cT3 ≈ cT4; matching degree 0.85) [24]. Currently, fuzzy matching can either be performed via traditional metric algorithms (e.g., Levenshtein Distance for string data “tumor site”) or by means of artificial intelligence. The latter may achieve optimal matching accuracy via fuzzy interference algorithms at the cost of requiring extensive data sets of up to 10,000 data pairs [25]. It is understood that traditional fuzzy matching performed in the present study carries a high risk of bias [26]. However, we could not imagine a viable alternative to comparing these two groups of patients. Propensity score matching resulted in significantly less comparable case and control groups (data not shown) than fuzzy matching, which generated an acceptable covariate balance (Table 3). Applying artificial intelligence-based algorithms was omitted based on the comparably small number of data sets available.

One non-surgically treated control patient had a single lung metastasis. This patient received additional stereotactic body irradiation and survived with good functional outcome. The surgical group contained more tumors of the oral cavity as the ITB preferred surgical treatment of this tumor site, but this is not considered to cause relevant bias. Overall, the survival of the 21 downstaged cases with treatment de-escalation and matched controls with primary RT or ST/RT did not differ (*p* = 0.18). However, more patients in the case group reported better values for the functional domain nutrition (*p* = 0.002). Thus, patients who have a high likelihood of lymph node downstaging and treatment de-escalation may benefit from receiving primary surgical treatment. This is the case in about 15% of cN+ patients. These are almost exclusively patients with cT1–2 tumors with clinically positive cervical lymph nodes of stages cN1 and cN2b. Typically, cervical lymph nodes do not have central necrosis and have a maximum short axis diameter below 12 mm (Table 2). In these patients, a selective therapeutic ND is frequently sufficient [2]. If patients are not well suited for a primary surgical procedure, PET-CT should be performed. If lymph nodes are PET negative, performing a primary RT without systemic therapy can be considered. However, this procedure leaves a certain diagnostic uncertainty. As long as no clear results from de-escalation studies are available, this also applies to p16-positive oropharyngeal tumors. It is understood that further investigation is needed to substantiate these observations, but it is believed that the problem of clinical overstaging in head and neck cancer deserves more attention.

## 5. Conclusions

Cervical lymph node overstaging is an underestimated problem in head and neck cancer. It may occur in about 25% of patients clinically staged as neck node positive. Subsequent overtreatment may cause preventable harm. In HNC patients with cT1–2 tumors with cN1 or cN2b neck lymph nodes, surgical treatment offers a chance to recognize overstaging and de-escalate treatment. If patients are not suitable for surgical treatment, a PET-CT may provide information as to whether RT without systemic therapy is sufficient.

## Figures and Tables

**Figure 1 diagnostics-12-01377-f001:**
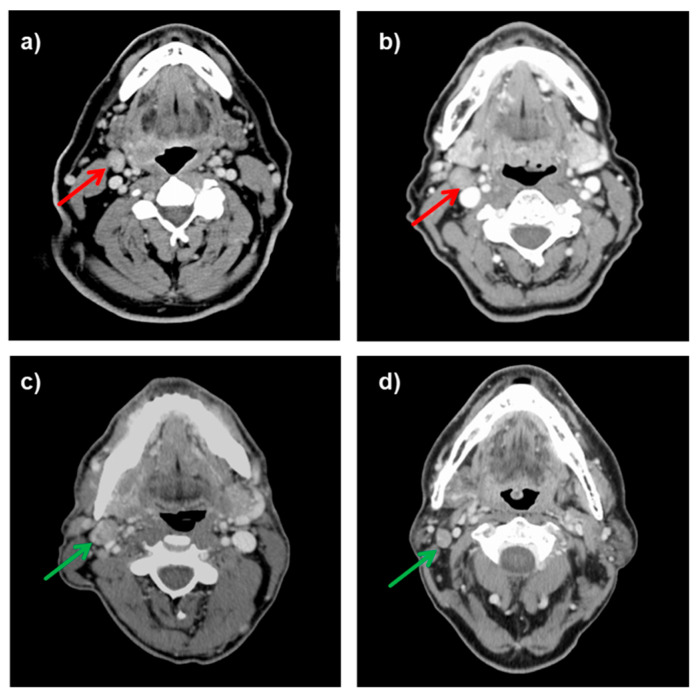
Axial contrast-enhanced CT scans of HNC patients clinically staged as neck lymph node positive who were histopathologically negative and who could be rejected from postoperative irradiation (upper row). In the lower row, CT scans of patients who were actually histopathologically positive and who had postoperative irradiation (lower row). Red arrow indicates cervical lymph node overstaging. Green arrow indicates true positive lymph node staging: (**a**) A 62-year-old male patient with a cT2cN2bcM0 oropharyngeal cancer treated with lateral pharyngotomy and neck dissection; (**b**) A 61-year-old female patient with a cT2cN2bcM0 supraglottic laryngeal cancer treated with transoral resection and neck dissection; (**c**) A 54-year-old male patient with a cT2cN2bM0 (pT2pN1) oropharyngeal cancer treated with transoral resection, neck dissection and PORT; (**d**) A 57-year-old male patient with a cT2cN2cM0 (pT2pN2c) supraglottic laryngeal cancer treated with transoral resection, bilateral neck dissection and PORT.

**Figure 2 diagnostics-12-01377-f002:**
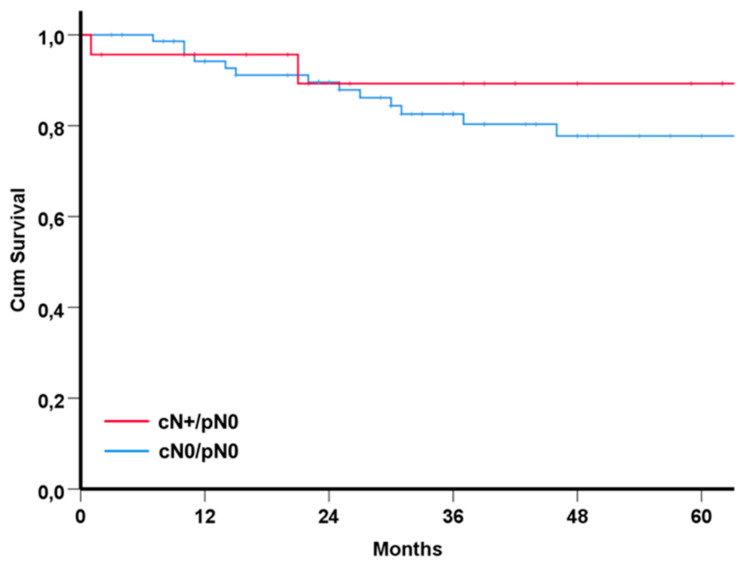
Kaplan–Meier plots of 24 patients with originally planned surgical tumor resection, therapeutic neck dissection and PORT (red line), in whom treatment was finally de-escalated to surgery alone (cases), and patients staged as cN0 (blue line) who were a priori scheduled for primary surgical resection with elective neck dissection only (controls; *n* = 72; log rank *p* = 0.74).

**Figure 3 diagnostics-12-01377-f003:**
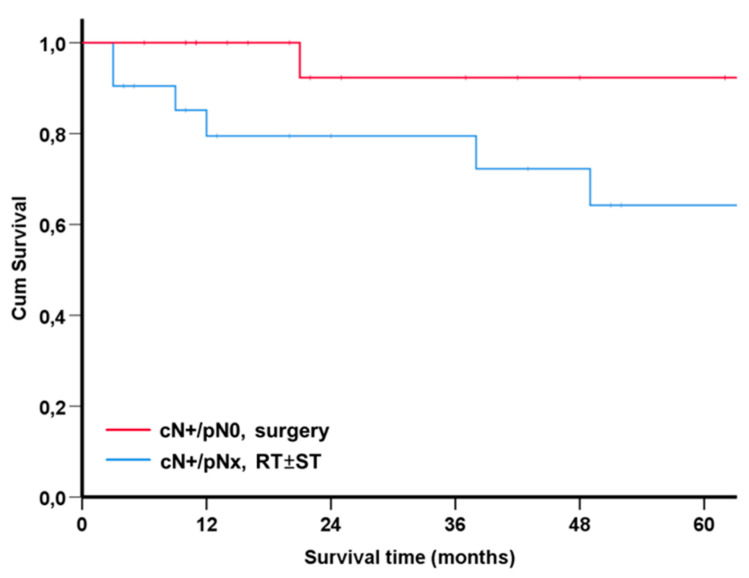
Kaplan–Meier plots of 21 patients with originally planned surgical tumor resection, therapeutic neck dissection and PORT (red line), in whom treatment was finally de-escalated to surgery alone (cases), and matching patients with cN+ (blue line) who were admitted to definitive primary RT ± ST (controls; *n* = 21; log rank *p* = 0.18).

**Table 1 diagnostics-12-01377-t001:** Patient and disease characteristics of 292 patients with incident head and neck carcinoma, who received a ND as part of initial treatment. Clinical stage at time of diagnosis; prim. RT + ST: primary definitive radiotherapy with systemic therapy; prim. RT: primary definitive radiotherapy.

Variable	Attribute	Count	Percent
Sex	Male	227	78%
	Female	65	22%
Age at first diagnosis	≤50	46	16%
	51–60	97	33%
	61–70	86	29%
	71–80	51	17%
	>80	12	4%
ASA I/II vs. ASA III/IV	ASA I/II	132	63%
	ASA III/IV	78	37%
Common tumor sites	Lips and oral cavity	63	22%
	Oropharynx	117	40%
	Hypopharynx	16	5%
	Larynx	60	21%
	Others	36	12%
cT stage	T1	75	26%
	T1a	3	1%
	T1b	1	0%
	T2	127	44%
	T3	47	16%
	T4	6	2%
	T4a	30	10%
	T4b	3	1%
cN stage	N0	128	44%
	N1	57	20%
	N2a	10	3%
	N2b	77	26%
	N2c	20	7%
Clinical UICC Stage	Stage I	47	16%
	Stage II	54	18%
	Stage III	66	23%
	Stage IVa	120	41%
	Stage IVb	3	1%
	Stage IVc	2	1%
Treatment modalities	Surgery only	120	41%
	Surgery & PORT	119	41%
	Surgery & RT + S7	53	18%
	prim. RT + ST	0	0%
	prim. RT	0	0%

**Table 2 diagnostics-12-01377-t002:** Characteristics of cN+/pN0 vs. cN+/pN+ cervical lymph nodes. Characteristics in clinically LN-positive patients by histopathological outcome of neck specimen and chi-square *p*-values. Increasing the threshold for clinical lymph node positivity from a short axis diameter of at least 10 mm to at least 15 mm would reduce specificity without better sensitivity.

		pN Negative		pN Positive		Total		*p*-Value
		Count	Column N %	Count	Column N %	Count	Column N %	
Common tumor sites	Lips and oral Cavity	11	26%	19	15%	30	18%	0.21
	Oropharynx	21	49%	61	50%	82	50%	
	Hypopharynx	0	0%	9	7%	9	5%	
	Larynx	9	21%	21	17%	30	18%	
	Others	2	5%	11	9%	13	8%	
	Total	43	100%	121	100%	164	100%	
T stage truncated	T1	7	16%	25	21%	32	20%	0.51
	T2	22	51%	51	42%	73	45%	
	T3	9	21%	21	17%	30	18%	
	T4	5	12%	24	20%	29	18%	
	Total	43	100%	121	100%	164	100%	
cN at initial diagnosis	N1	24	56%	33	27%	57	35%	0.007
	N2a	2	5%	8	7%	10	6%	
	N2b	12	28%	65	54%	77	47%	
	N2c	5	12%	15	12%	20	12%	
	Total	43	100%	121	100%	164	100%	
Lymph node necrosis	0	41	95%	86	74%	127	80%	0.003
	1	2	5%	30	26%	32	20%	
	Total	43	100%	41	100%	159	100%	
CT short axis diameter grouped (mm)	≤5	0	0%	1	1%	1	1%	0.001
	6–10	19	44%	18	16%	37	23%	
	11–15	19	44%	41	35%	60	38%	
	16–20	4	9%	17	15%	21	13%	
	21–25	1	2%	21	18%	22	14%	
	26–30	0	0%	13	11%	13	8%	
	31–35	0	0%	3	3%	3	2%	
	36+	0	0%	2	2%	2	1%	
	Total	43	100%	116	100%	159	100%	

**Table 3 diagnostics-12-01377-t003:** Case control matching of surgically treated patients with treatment de-escalation (cases) and patients with primary RT ± ST (controls). Using case control matching, we identified 21 matched pairs, i.e., 21 surgically treated patients with treatment de-escalation (cases) and 21 patients with primary RT ± ST (controls). Matching variables included gender, tumor site, cT-stage, cN-stage and p16 positivity in patients with oropharyngeal carcinoma, and ASA score as a comorbidity indicator. For three surgically treated patients, no appropriate matches could be identified.

Variable	Attributes	Cases	Controls	*p*-Value
Sex	Male	16	16	1.0
	Female	5	5	
ASA I/II vs. ASA III/IV	ASA I/II	15	12	0.33
	ASA III/IV	6	9	
Age groups at first diagnosis	≤50	1	1	0.7
	51–60	8	4	
	61–70	9	11	
	71–80	2	4	
	>80	1	1	
P16 (oropharynx only)	p16 negative incl. oropharynx	20	20	1.0
	p16 positive oropharynx	1	1	
Common tumor sites	Lips and oral cavity	8	3	0.24
	Oropharynx	9	9	
	Hypopharynx	1	2	
	Larynx	3	7	
	Others	0	0	
cT at initial diagnosis	T1	6	3	0.14
	T2	13	11	
	T3	2	7	
cN at initial diagnosis	N0	0	5	0.11
	N1	13	7	
	N2a	1	2	
	N2b	6	5	
	N2c	1	2	
Clinical UICC-stage	Stage 1	0	1	0.17
	Stage 2	0	3	
	Stage 3	13	8	
	Stage 4a	7	9	
	Stage 4b	0	0	
	Stage 4c	1	0	

**Table 4 diagnostics-12-01377-t004:** Head and neck functional integrity scale outcome of surgically treated patients with treatment de-escalation (cases) and patients with primary RT ± ST (controls).

Functional Domain	Integrity Grade	Case	Control	*p*-Value
Nutrition	Unable to swallow; only via gastrostomy tube	0	1	0.002
	Via gastrostomy tube and oral	0	3	
	No gastrostomy tube, oral diet, but only liquid/soft food	0	3	
	No gastrostomy tube, diet slightly restricted	3	4	
	Normal	14	6	
Breathing	Tracheostoma, blocked cannula	0	0	1.0
	Tracheostoma, speech cannula/no cannula	3	1	
	No tracheostoma, breathing difficulties at rest	0	0	
	No tracheostoma, breathing difficulties only on exertion	1	4	
	Normal	13	12	
Speech	Not possible without phonation	0	0	0.44
	Difficult to understand, no phone calls	1	1	
	Telephoning possible	0	0	
	Easy to understand, but pronunciation/voice changed	4	7	
	Normal	12	9	
Pain	Pain despite opiate therapy	0	1	0.46
	Needs opiates	1	0	
	Regularly needs non-opioid analgesics	0	1	
	Needs analgesics from time to time	1	2	
	No pain	15	13	
Mood	Suicidal thoughts	0	0	1.0
	Very depressed despite antidepressants	0	0	
	With antidepressants overall normal mood	0	1	
	Occasionally depressed, no antidepressants needed	2	1	
	Normal	15	15	
Neck and shoulder mobility ^1^	Stiff neck, hardly any movement possible	0	0	0.42
	Can hair hardly comb, looking backwards in car not possible	0	1	
	Combing with problems, looking backwards in car difficult	1	2	
	Combing and looking backwards in car slightly restricted	3	3	
	Normal	13	11	

^1^ The worse result of neck mobility and shoulder mobility is counted.

## Data Availability

The data presented in this study are available on request from the corresponding author.

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
