# Peer review of "Frequency and Consequences of Cervical Lymph Node Overstaging in Head and Neck Carcinoma"

_diagnostics, 2022, doi:10.3390/diagnostics12061377_

Round 1

Reviewer 1 Report

The topic is very important as patients with Head and Neck carcinoma have unfortunate outcomes and provide less time for extensive invasive therapy. Here are my comments to reshape the manuscript before considering it for publication: 

  • After the abstract, the authors only started talking about the fuzzy Matching Algorithm. However, I would recommend adding a section on it and all available algorithms and machine learning techniques for such studies. It can be a few lines but should be mentioned in the literature review.
  • In the discussion, please write a section on the strengths and weaknesses of the current study.

Author Response

Enclosed, please find our revised manuscript “Frequency and consequences of cervical lymph node overstaging in head and neck carcinoma” for possible publication in “Diagnostics”.

We appreciate the thoughtful review of our paper. We have changed and modified the manuscript accordingly. We shall address now the reviewer’s suggestions point by point:

Answers to comments of Reviewer 1:

The authors thank the reviewer for the helpful comments:

„The topic is very important as patients with Head and Neck carcinoma have unfortunate outcomes and provide less time for extensive invasive therapy. Here are my comments to reshape the manuscript before considering it for publication: 

  • After the abstract, the authors only started talking about the fuzzy Matching Algorithm. However, I would recommend adding a section on it and all available algorithms and machine learning techniques for such studies. It can be a few lines but should be mentioned in the literature review.“

The authors very much appreciate this valuable suggestion. However, we believe it is impossible to include all available machine learning techniques for fuzzy matching without severely compromising the integrity of the current manuscript.

Instead, we included a concise paragraph in the Discussion section, explaining the basic principle of fuzzy matching. Thus, frequently used traditional algorithms (e.g. Levenshtein Distance) and artificial intelligence based algorithms (e.g. fuzzy interference) were briefly introduced. In addition, we highlighted the reason for choosing a tradition fuzzy matching approach in more detail. In short, this choice was mainly based on the extensive number of data sets required for optimal matching accuracy if artificial intelligence based fuzzy matching would have been applied.

Please refer to the Discussion section of the revised manuscript, page 11, paragraph 3.

  • „In the discussion, please write a section on the strengths and weaknesses of the current study.“

We thank the reviewer for this suggestion. We have added a section in the revised manuscript on page 11, paragraph 2.

Reviewer 2 Report

This manuscript reports the possible risk of overdiagnosis and overtreatment for cervical lymph node status of head and neck carcinoma. Many surgeons of OMS are probably aware of the risk of overtreatment, however, it has not been reported with evidence based on a certain number of patients. I think this manuscript will provide new insights into diagnosis which will lead to sophistication of treatment strategy.

Someone thinks that this can be accepted without any correction, however, I’d like to offer some minor considerations especially for readers in non-European countries where the treatment strategy (including devices for diagnosis and treatment) is quite different.

Minor point:

  1. In lines 51-60, there are four questions. Each question will be clearly visualized and understood if the line would be changed.
  2. Two groups compared in this manuscript, patients with treatment de-escalation and patients with primary RT may be difficult to understand for readers in non-European countries. View of life is totally different between Western countries and Asian/Middle-east countries which may affect the treatment strategy. Moreover, a set of CT scanning, PET-CT, and needle biopsy is completely performed in some Asian countries, and their first treatment strategy is surgical treatment. Therefore, I recommend you clarify the definition of the two compared groups with easy words.

Author Response

Enclosed, please find our revised manuscript “Frequency and consequences of cervical lymph node overstaging in head and neck carcinoma” for possible publication in “Diagnostics”.

We appreciate the thoughtful review of our paper. We have changed and modified the manuscript accordingly. We shall address now the reviewer’s suggestions point by point:

Answers to comments of Reviewer 2:

This manuscript reports the possible risk of overdiagnosis and overtreatment for cervical lymph node status of head and neck carcinoma. Many surgeons of OMS are probably aware of the risk of overtreatment, however, it has not been reported with evidence based on a certain number of patients. I think this manuscript will provide new insights into diagnosis which will lead to sophistication of treatment strategy.

Someone thinks that this can be accepted without any correction, however, I’d like to offer some minor considerations especially for readers in non-European countries where the treatment strategy (including devices for diagnosis and treatment) is quite different.

Minor point:

  • In lines 51-60, there are four questions. Each question will be clearly visualized and understood if the line would be changed.

We thank the reviewer for this suggestion. We have changed accordingly in the revised manuscript.

  • Two groups compared in this manuscript, patients with treatment de-escalation and patients with primary RT may be difficult to understand for readers in non-European countries. View of life is totally different between Western countries and Asian/Middle-east countries which may affect the treatment strategy. Moreover, a set of CT scanning, PET-CT, and needle biopsy is completely performed in some Asian countries, and their first treatment strategy is surgical treatment. Therefore, I recommend you clarify the definition of the two compared groups with easy words.

We thank the reviewer for this interesting comment. We have addressed this point in the revised manuscript on page 7, paragraph 4 and page 11, paragraph 2.
